# A Single-Cell Atlas of Porcine Skeletal Muscle Reveals Mechanisms That Regulate Intramuscular Adipogenesis

**DOI:** 10.3390/ijms252312935

**Published:** 2024-12-01

**Authors:** Zhong Xu, Junjing Wu, Yujie Li, Jiawei Zhou, Yu Zhang, Mu Qiao, Yue Feng, Hua Sun, Zipeng Li, Lianghua Li, Favour Oluwapelumi Oyelami, Xianwen Peng, Shuqi Mei

**Affiliations:** 1Hubei Key Laboratory of Animal Embryo and Molecular Breeding, Institute of Animal Husbandry and Veterinary, Hubei Provincial Academy of Agricultural Sciences, Wuhan 430064, China; xz8907@163.com (Z.X.); jeanne1106@126.com (J.W.); liyujie13653814659@163.com (Y.L.); zhoujiawei@hbaas.com (J.Z.); zhangyu@hbaas.com (Y.Z.); mqbetter@163.com (M.Q.); fy914858517@163.com (Y.F.); sunhua--1975@163.com (H.S.); lizipeng@hbaas.com (Z.L.); lilianghua@hbaas.com (L.L.); 2The John Curtin School of Medical Research, Australian National University, Canberra, ACT 2601, Australia; oyefavour@gmail.com

**Keywords:** pig, snRNA-seq, intramuscular fat, cell cluster, adipogenesis

## Abstract

Porcine skeletal muscle development is closely linked to meat production efficiency and quality. The accumulation of porcine intramuscular fat is influenced by the hyperplasia and hypertrophy of adipocytes within the muscle. However, the cellular profiles corresponding to the two stages of muscle development remain undetermined. Single-nucleus RNA sequencing (snRNA-seq) can elucidate cell subsets in tissues, capture gene expression at the individual cell level, and provide innovative perspectives for studying muscle and intramuscular fat formation. In this study, a total of 78,302 nuclei and 9 clusters of cells, which included fibro/adipogenic progenitor (FAP), myonuclei, adipocytes, and other cell types, of Xidu black pigs, were identified on Day 1 and Day 180. The pattern of cell clustering varied between the two developmental stages. Notably, the percentage of adipocytes in the Day 180 group was higher than in the Day 1 group (0.51% vs. 0.15%). Pseudo-time sequence analysis indicated that FAPs could differentiate into adipocytes and myonuclei cells, respectively. The *THRSP* gene was identified as a biomarker for swine intramuscular fat cells, and its down-regulation resulted in significant reduction in lipid droplet formation in porcine preadipocytes. Our research provides new insights into the cellular characteristics of intramuscular fat formation, which may facilitate the development of novel strategies to enhance intramuscular fat deposition and improve pork quality.

## 1. Introduction

As living standards improve, the demand for higher-quality pork increases [1]. The intramuscular fat (IMF) content is an important factor that affects pork quality [2]; thus, optimizing the IMF content is a vital target in pig breeding operations. At present, the key genes and molecular mechanisms that affect IMF deposition are unclear, and this is limiting the breeding efficiency of pork quality improvement [3]. Therefore, it is critical to identify the key genes and regulatory mechanisms of porcine IMF deposition.

Skeletal muscle development depends on the two processes of myogenesis and adipogenesis. Both myocytes and adipocytes are derived from mesenchymal progenitor cells [4,5], and their development is divided into two stages: the determination stage (hyperplasia) and the terminal differentiation stage (hypertrophy) [6]. Hyperplasia mainly involves increasing the number of cells. Porcine skeletal muscle cells are generally in the hyperplastic stage from the late embryonic to the early postnatal period. In the later developmental phases, the cells mainly increase in size and enter the hypertrophic stage. Around 180 days after birth, the most obvious changes in IMF deposition occur [7]. At this time, the skeletal muscle cells undergo hypertrophy, the volume of fat cells increases significantly, and they continue to differentiate and deposit fat. Therefore, it is ideal to use pigs at different developmental stages to study fat deposition.

IMF is distributed between muscle bundles, and it is very difficult to isolate IMF directly from muscle tissue [8]. In addition, bulk RNA sequencing of muscle-related adipocytes is challenging due to interference from other cell types, such as muscle cells [9]. Single-cell RNA sequencing (scRNA-seq) technology is now being used to determine the gene expression status of single cells, and it provides a new opportunity to study muscle-related adipocytes that are difficult to extract in large quantities [10]. For example, scRNA-seq has been used to confirm that the difference in fat deposition between lean and obese pigs may be related to the Ca^2+^ concentration in the cytoplasm and endoplasmic reticulum of muscle cells [11]. In addition, single-nucleus RNA sequencing (snRNA-seq) is a high-resolution transcriptome method that can be used to assess both mononuclear and multinucleated cells [12], which helps overcome the challenges associated with studying the large myotubes and myofibers of skeletal muscle. However, to date, no study has been conducted on pigs at different stages of development using snRNA-seq.

Xidu black pigs have a higher IMF content than lean pig breeds, and they can serve as a good model for studying IMF deposition [13]. In this study, snRNA-seq was performed to elucidate the diversity of the porcine *longissimus dorsi* muscle (LDM) and identify IMF marker genes. The heterogeneity and cellular composition of the porcine LDM, as well as the molecular characteristics of the muscle- and IMF-producing cells, were analyzed at two different developmental stages. Our findings provide new insights into myogenesis and intramuscular adipogenesis.

## 2. Results

### 2.1. SnRNA-Seq Identified Distinct Cell Populations in Longissimus Dorsi Muscle at Two Developmental Stages

To study the mechanism of development of the porcine LDM, cells were extracted from LDM tissue on Day 1 (D1) and Day 180 (D180) using the 10× Genomics Chromium platform (Figure 1A). According to the Cell Ranger analyses, the estimated number of nuclei was 78,302. The average reads, median genes, and median unique molecular identifier (UMI) counts per nucleus were also obtained (Table 1). After controlling for the quality of the snRNA-seq data, 63,271 nuclei from four separate libraries (43,586 nuclei from two D1 libraries and 19,685 nuclei from two D180 libraries) were retained for downstream analysis (Table 2). Using Seurat (v4.0.4), the aggregated and normalized snRNA-seq data were clustered to identify the associated cell types, as shown in the t-distributed stochastic neighbor embedding (t-SNE) diagram presented in Figure 1B. Based on the expression of lineage-specific markers, we identified the following nine cell types: muscle fiber cells (*NOS1*, *TACC2*), tenocytes (*MKX*), smooth muscle cells (*GUCY1A2*), immune cells (*PTPRC*), FAPs (*PDGFRA*), endothelial cells (*PECAM1*), satellite cells (*PAX7*), adipocytes (*ADIPOQ*), and Schwann cells (*CDH19*) (Figure 1C).

A comparative analysis of the D1 and D180 LDM cell populations showed that there were higher percentages of myonuclei cells and immune cells in the D180 group compared to the D1 group (73.66% vs. 52.01%, 3.65% vs. 0.70%), and lower percentages of satellite cells, tenocytes, and FAPs (2.99% vs. 5.45%, 3.32% vs. 12.44%, 9.69% vs. 22.21%). Notably, the percentage of adipocytes was higher in the D180 group compared to the D1 group (0.51% vs. 0.15%). Endothelial cells, smooth muscle cells, and Schwann cells were all found to account for slightly higher proportions in the D1 group compared to the D180 group (Figure 1D).

### 2.2. Clustering Analysis Identified Subgroups in the Subpopulation of Myonuclei Cells

To study the cellular mechanism underlying IMF deposition in Xidu black pigs, we performed a subpopulation analysis using the D1 and D180 LDM samples and determined that the adipocyte, muscle cell, and FAP subpopulations were the key cell subpopulations. Among these subpopulations, myonuclei cells accounted for the largest proportion of LDM samples from pigs of different ages (Figure 1D), and they play an important role in muscle tissue development.

We performed an unsupervised cluster analysis using identified marker genes, and a total of five cell subgroups were noted (Figure 2A,B): type I myonuclei cells (*MYH7*), type IIb myonuclei cells (MYH4), myotendinous junction (MTJ) cells (*COL6A3*), type IIx myonuclei cells (*MYH1*), and neuromuscular junction (NTJ) cells (*VAV3*). MTJ cells were previously found to interact with satellite cells, and NTJ cells were previously found to improve satellite cell function [14]. In addition, it has been shown that cells expressing the myonuclear marker gene *NOS1* interact with satellite cells [15]. *MYH4* and *MYH1* have been reported as marker genes for fast muscle fibers in mice and humans, and *MYH7* is usually a marker gene for slow muscle fibers. When we examined the proportion of each cell subtype in the samples from the D1 and D180 pigs, we found that type IIb myonuclei cells accounted for the largest proportion in the D180 samples and that type I myonuclei cells accounted for the largest proportion in the D1 samples (Figure 2C). These results indicate that during the earlier developmental phases, porcine muscle is rich in slow muscle fibers, while in the later stages of development, it is mainly rich in fast muscle fibers. We checked some of the literature [16], which is consistent with our results. The researchers found that the number of type I slow muscle fiber is greater in Chinese Lantang and Landrace pigs at birth, while the number of type IIB fast muscle fibers is greater at 90 days postpartum.

Next, we conducted a GO analysis of the genes that were upregulated in these two cell subgroups. An examination of the top 30 enriched biological processes showed that the type IIb myonuclei fast muscle fibers were mainly involved in protein modification, various metabolic regulatory processes, muscle structure development, muscle cell differentiation, and cytoskeletal organization processes (Figure 2D). The type I myonuclei slow muscle fibers were found to be mainly involved in oxidative phosphorylation, ATP metabolism, nucleotide metabolism, mitochondrial ATP synthesis coupling, respiratory electron transport chain, and other biological processes (Figure 2E). Taken together, these results provide evidence of the cellular composition, specific marker genes, and biological processes of the LDM of Xidu black pigs at the D1 and D180 time points.

### 2.3. Analysis of the Heterogeneity of the FAP Subpopulation via Unsupervised Clustering

To investigate the cellular origin of IMF, we analyzed the FAP subpopulation. FAPs can reportedly differentiate into adipocytes and fibroblasts, and this characteristic has important implications for pork quality. To explore the heterogeneity of the FAP subpopulation, FAPs were separated into seven clusters: *POSTN*_FAPs_0, *LPL*_FAPs_1, *TNMD*_FAPs_2, *VCAM1*_FAPs_3, *MLIP*_FAPs_4, *IGF2R*_FAPs_5, *NR2F2*_FAPs_6 (Figure 3A,C). It was found that the cells within Cluster FAPs 0 and FAPs 6 were present in greater proportions in the D180 group compared to the D1 group (Figure 3B).

The cells within Cluster FAPs 0 were enriched for GOBP terms related to growth factors, motility, and cell migration regulation (Figure 3D). In addition, while the cells within Cluster FAPs 6 were associated with non-classical Wnt/planar cell polarity signaling (Figure 3E), cells within Cluster FAPs 0 were mainly associated with the classical Wnt signaling pathway, including the expression of Wnt-related genes (e.g., *GPC3*). It is worth noting that Cluster FAPs 6 cells were also associated with the circadian rhythm, which has been reported to interact with lipid metabolism, liver gluconeogenesis, and insulin resistance [17,18]. The genes within Cluster FAPs 0 were also found to be enriched for transmembrane receptor protein serine/threonine kinase-signaling pathway processes. These pathways are associated with lipogenesis and lipid metabolism [19].

In addition, we found that the Cluster FAPs 4 cells were significantly enriched for biological processes involved in myocyte development, muscle development and differentiation, actin-filament-based process, myofibrillar aggregation, cytoskeleton organization, actin cytoskeleton organization, myocardial development, and heart development (Figure 3F). We also found that the Cluster FAPs 2 cells performed functions related to the following structures: the extracellular matrix, collagen (catabolism), collagen fiber tissue, supramolecular fibers, fibril tissue, the endoplasmic reticulum, and contractile fibers (Figure 3G). Therefore, Cluster FAPs 2 and Cluster FAPs 4 cells may be involved in fibroblast formation and muscle formation.

The GOBP enrichment analysis of the cells in Cluster FAPs 5 showed that they were associated with development, positive transcriptional regulation, metabolism, muscle tissue development, and striated muscle development (Appendix A).

In conclusion, in this section of the study, we analyzed the FAPs found in the LDM of Xidu black pigs, divided them into seven clusters, and defined the function of each cluster of cells.

### 2.4. Genes Commonly Expressed During Two Stages of Adipocyte Development

Analysis of the adipocyte populations in the D1 and D180 groups revealed 3691 differentially expressed genes: 774 upregulated genes and 2917 downregulated genes (Figure 4A). Lipogenesis-related genes (*CEBPA*, *PPARG*), mature fat marker genes (*SCD*, *LPL*), and lipid (*LIPE*, *PLIN1*) and fatty acid (*ACAA1*) metabolism-related genes were significantly upregulated in the D180 group (Figure 4B). Among these genes, the *THRSP* gene was significantly upregulated in the D180 adipocytes compared to the expression in the D1 adipocytes, which was consistent with the results of our previous ATAC-seq and RNA-seq studies [13]. Based on these results, we speculated that the *THRSP* gene plays a positive role in porcine fat deposition and thus selected this gene for in vitro functional verification.

As shown in Figure 4C, the GO enrichment analysis of the D1 and D180 adipocyte populations revealed 66 GO terms. The genes differentially expressed in the D1 and D180 cells were significantly enriched in biological processes such as cell, protein, and nitrogen compound metabolism. This indicated that considerable levels of amino acid synthesis and utilization occurred during adipose tissue growth. We identified genes involved in the positive regulation of adipocyte differentiation and proliferation, such as peroxisome proliferator activated receptor delta (*PPARD*), CCAAT enhancer binding protein alpha (*CEBPA*), peroxisome proliferator activated receptor gamma (*PPARγ*), and fatty acid binding protein 4 (*FABP4*).

The KEGG pathway enrichment analysis showed that the genes differentially expressed in the adipocytes were enriched in Alzheimer’s disease, non-alcoholic fatty liver disease, insulin signaling, fatty acid metabolism, and other pathways, suggesting that these cells may play an important role in fat deposition (Figure 4D).

To confirm the role that the *THRSP* gene plays in lipid droplet formation at the molecular level, we knocked down the *THRSP* gene in porcine preadipocytes using siRNA. We determined the effect of the *THRSP* gene knockdown on adipogenic marker genes and indicated that the mRNA expression of the genes was markedly reduced in the *THRSP*-knockdown group compared to the control group (Figure 4E). Knocking down the *THRSP* gene resulted in a significant reduction in lipid droplet formation in preadipocytes (Figure 4F).

### 2.5. Pseudotime Analysis Revealed Key Progenitor Cell Characteristics and Gene Sets of FAPs

A pseudotemporal analysis of FAPs, adipocytes, and myocytes was performed using Monocle 2 (version 2.12.0, written by the Trapnell Lab, Department of Genome Sciences, University of Washington). As shown in Figure 5A, the FAPs had two differentiation trajectories. At branch point 1, the trajectories pointed toward differentiation into adipocytes and myonuclei cells, confirming the bidirectional differentiation ability of FAPs. To determine and compare the gene expression in the cells in each branch, five gene sets were identified and utilized, based on the gene expression in cells with different differentiation fates (Figure 5B,C). States 1 and 2 marked the differentiation of FAPs into adipocytes. States 1 and 3 marked the differentiation of FAPs into myocytes (Figure 5C). Cluster 1 and 2 genes (related to adipocyte differentiation) were significantly associated with *COL4A1, GPC6*, *MAML2*, *ELOVL6*, and *PDE3A*. *SCARA5* was mainly expressed in FAPs, and its expression gradually decreased with differentiation. At the branch points, increased expression was observed at the end of states 1 and 2, and increased expression was also observed in the myonuclei cells during differentiation. The marker genes for the myonuclei cells (*TACC2* and *NOS1*) were only expressed in small quantities of FAPs at the early stage of differentiation; however, they were mainly expressed in the myonuclei cells from the middle stage of differentiation, and the expression increased significantly during the middle and late stages of differentiation states 1 and 3. The expression of *ADIPOQ* significantly increased in the middle and late stages of differentiation states 1 and 2, and it was almost exclusively expressed in adipocytes in the pseudo-time locus, with a slight downward trend only during the middle stages of differentiation.

We then conducted a GO enrichment analysis of the differentially expressed genes (Figure 5D). *SOX5*, *GPC6*, *IGF1*, *CFD*, *DCN*, *LPL*, and other genes in cluster 1 showed specific high expression at the end of the branch of cell fate 2 and at branch point 1, while *ADIPOQ* and *ELOVL6* were very weakly expressed at branch point 1; however, they were only highly expressed at the terminal of cell fate 2. The GO enrichment of the cluster 1 genes showed that they were related to extracellular regions, the cytoplasm, and the extracellular matrix. Similarly, the *PDE3A*, *MAML2*, *COL4A1*, *SCARA5*, and *COL6A3* genes in cluster 2 were expressed basically to the same extent as those in cluster 1, and they were all highly expressed at the branch point and the end of the branch of cell fate 2. The GO enrichment results of this cluster were also similar to those of cluster 1 (e.g., association with the extracellular matrix); however, the cluster 2 genes also demonstrated associations with cell migration, motility, and extracellular structural tissues. In contrast, the genes in cluster 3, such as *TRDN*, *TTN*, *NEB*, *CMSS1*, *DMD*, *NOS1*, and *TACC2*, showed an opposite trend to those in clusters 1 and 2. The expression of these genes at the end of the branch of cell fate 3 was significantly higher than at the branch point. The GO enrichment of this cluster of genes showed that they were related to myofibrils, contractile fibers, and the I band of striated muscle. The genes in cluster 4 were associated with muscle system processes, muscle contraction, myofilms, and myofibrils in the GO enrichment results, demonstrating similar associations to the cluster 3 genes. *MYH4*, *TXLNB*, *EPM2A*, *GRIK2*, *KCNQ5*, *TP63*, *TOMD1*, *MLIP*, and other genes were minimally expressed at the branch site and cell fate 2; however, their expression increased in the middle part of the branch of cell fate 2. The expression of these genes gradually increased during the middle and late stages of the branch of cell fate 3 (Figure 5B,D).

Together, these results show that the differentiation relationships among FAPs, myonuclei cells, and adipocytes and the functional characteristics of the different gene sets provide a theoretical basis for the bidirectional progenitor characteristic of FAPs.

## 3. Discussion

Given that pork is the most widely consumed type of meat worldwide, it is not surprising that it is the subject of considerable research to optimize its quality and IMF content. However, exploring potential functional differences and transcriptional heterogeneity in skeletal muscle is technically challenging since it contains both multinucleated (e.g., myofibers) and mononuclear (e.g., fat cells) components [9]. SnRNA-seq is a novel high-resolution transcriptomic method that can be used to detect transcriptome mononuclear and multinucleated cells simultaneously [12]. In this study, we used snRNA-seq to construct a cell map of the LDM of Xidu black pigs at different developmental stages. We also analyzed the heterogeneity of the LDM’s myonuclei cells, FAPs, and adipocytes, including via quasi-temporal trajectory analysis. This marks the first time that the heterogeneity of the LDM’s cell population has been studied at different ages in Xidu black pigs at the single-cell transcriptome level.

In previous studies, scRNA-seq has been used to analyze the LDM of different pig breeds [9,11,20]. In our study, the proportions of myonuclei cells and FAPs in the LDM were much higher than those of other cell types, and this finding differs from those of studies of cell population patterns in wild boar and Duroc, Laiwu, and Suhuai pigs. Our results suggest that Xidu black pigs have high research value in terms of the study of the development of muscle tissue and adipose tissue.

According to the expression of marker genes, three subsets of myonuclei cells were identified: type I myonuclei cells (*MYH7*), type IIb myonuclei cells (*MYH4*), and type IIx myonuclei cells (*MYH1*). The proportions of the different cell subsets indicated that there were more fast muscle fibers present at D180 and more slow muscle fibers present at D1 in the LDM. In addition, previous studies have shown that *MYH4* expression is significantly positively correlated with muscle fiber diameter [21]. This implies that the muscle fiber diameter was significantly larger at D180 compared to D1, which is consistent with the growth and development of pigs.

Mammalian adipose tissue is derived from mesenchymal stem cells (MSCs). Recent studies have shown that muscle-derived MSCs can be transformed into myogenic or FAPs. Subsequently, FAPs differentiate into adipocytes and fibroblasts in muscle tissue [22]. In this study, we found that the Cluster FAPs 4 cells were significantly enriched for biological processes involved in myocyte development and muscle development, and the Cluster FAPs 4 cells were enriched for biological processes involved in myocyte development. Therefore, Cluster FAPs 2 and Cluster FAPs 4 cells may be involved in fibroblast formation and muscle formation. *SERPINE1* and *LPL* gene expression was significantly higher in the Cluster FAPs 1 cells, and the *CFD*, *PPARG*, *VCAM1*, and *DLK1* genes were differentially expressed in the Cluster FAPs 3 cells. *LPL*, *VCAM1*, and *DLK1* are reportedly associated with committed preadipocytes, and *PPARG* and *SERPINE1* are known to be involved in fat deposition. Hence, we initially defined the cells in Clusters FAPs 1 and FAPs 3 as committed preadipocytes.

The pseudotime sequence analysis of the FAPs, myonuclei cells, and adipocytes showed that there was a branch node and two differentiation states that led separately toward the generation of myonuclei cells and adipocytes. Using five gene sets that represented the various differentiation fates, we identified differentially expressed genes that may regulate the differentiation fate of FAPs. Among them was the gene that encodes glypican 6 (*GPC6*); it is significantly related to IMF and growth traits, which in turn impact muscle development and fat deposition [23,24]. The mastermind-like transcriptional coactivator 2 (*MAML2*) gene was also identified; it is a milk cholesterol candidate gene and a key component of the Notch signaling and glycolide metabolism pathways [25]. The phosphodiesterase 3A (*PDE3A*) gene was also identified, and this protein plays an important role in the anti-lipolysis activity of adipocytes [26]. *TRDN*, which was significantly enriched in the branch associated with myonuclei cells, is a triplet protein and reportedly associated with excitation–contraction coupling and arrhythmia regulation in the myocardium [27]. However, *CMSS1*, which was also significantly enriched in the branch associated with myonuclei cells, was associated with abdominal fat deposition in broilers in a recent study [28], possibly due to differences between species. The results of the GO enrichment analysis which we performed using the five differential gene sets were consistent with the above analysis. In summary, these genes may regulate the direction of differentiation of FAPs, and their specific functions need to be further studied.

In a previous study, we identified the *THRSP* gene as a candidate marker for IMF deposition in the LDM using ATAC-seq and RNA-seq [13]. In this study, we found that the expression of the *THRSP* gene and some fat deposition marker genes (*PPARγ*, *FABP4*, *ADIPOQ*, *PLIN1*, *CEBPA*, *SCD1*, and *CIDEC*) positively correlated with IMF deposition in the adipocyte population. Here, we also determined that knocking down the *THRSP* gene resulted in a significant decrease in lipid droplet formation in porcine preadipocytes, suggesting that *THRSP* functions as a lipid-droplet-associated protein to regulate IMF deposition. At the same time, we noted that the *THRSP* gene was shown to be a marker of adipocyte number in the muscle of cattle and not of higher-fat-deposition activities [29], which may be similar in pig muscle.

## 4. Materials and Methods

### 4.1. Animals and Samples

Two *longissimus dorsi* muscle samples of Xidu black pigs were collected from neonatal (1 days) and adult (180 days) stages, respectively, from Hubei Tianzhili high-quality pig breeding Co., Ltd. (Enshi, Hubei, China). *Longissimus dorsi* muscle samples were collected and frozen in liquid nitrogen for subsequent experiments. All the experimental procedures were approved by the Institutional Animal Care and Use Committee of the Hubei Academy of Agriculture Sciences, and all the methods that involved pigs were in accordance with the agreement of the Institutional Animal Care and Use Committee of the Hubei Academy of Agriculture Sciences (permit number: 36/2016). The 3-day-old piglets used in the *THRSP* gene function validation test were obtained from the original pig breeding farm of the Institute of Animal Husbandry and Veterinary Medicine, Hubei Academy of Agricultural Sciences.

### 4.2. Library Construction and Sequencing

We cut the *longissimus dorsi* muscle sample into 2 mm pieces, and a Dounce homogenizer (885302-0002; Kimble Chase, Shanghai, China) was added into 2 mL cold nucleus EZ buffer and incubated on ice with additional 2 mL of lysis buffer for 5min. The homogenate was passed through a 40 mm cell strainer (43-50040-51; pluriSelect, Leipzig, Germany), and then centrifuged at 500× *g* at 4 °C for 5 min. The microspheres were resuspended, washed with 4 mL buffer, and incubated with ice for 5 min. After centrifugation, the microspheres were resuspended in a nuclei suspension buffer (PBS, 0.07% BSA, and 0.1% RNase inhibitor) and passed through a 20 μm cell strainer (43-50020-50; Complex selection) and calculated.

The library synthesis and RNA-sequencing were completed by a Gene Denovo (Guangzhou, China). In brief, the cells of each group were mixed into one sample and adjusted to 1000 cell/μL. The indexed sequencing libraries were prepared using Chromium Single Cell 3′ Reagent Kits (v2) according to the manufacturer’s instructions. The final Single Cell 3′ Libraries contained the P5 and P7 primers used in the Illumina bridge amplification PCR. The barcoded sequencing libraries were quantified using a standard curve-based qPCR assay (KAPA Biosystems, Wilmington, MA, USA) and Agilent Bioanalyzer 2100 (Agilent, Loveland, CO, USA). Finally, library sequencing was performed by Illumina HiSeq 4000 with a custom pair-end sequencing mode 26 bp (read 1) × 98 bp (read 2).

### 4.3. SnRNA-Seq Bioinformatics Analysis

The Cell Ranger Single Cell Software Suite (v6.1) was applied for quality control, and snRNA-seq data were aligned to Sus scrofa 11.1 as the reference genome. After the initial quality control, low-quality sequences with barcodes and unique molecular identifiers (UMIs) were removed. CellRanger version 2.0.0 and Seurat (v4.0.4) R package were used to filter out the low-quality cells. The following criteria were used to filter the cells: (1) gene counts >3000 per cell; (2) UMI counts >12,000 per cell; and (3) percentage of mitochondrial genes >15%. To visualize the data, we further reduced the dimensionality of all the cells using Seurat and used t-Distributed Stochastic Neighbor Embedding (tSNE) to project the cells into 2D space. We used the likelihood-ratio test to find differential expression for a single cluster, compared to all other cells. We identified differentially expressed genes by the following criteria: (1) Pvalue ≤ 0.01; (2) log2(fold change [FC]) ≥0.360674; and (3) the percentage of cells where the gene is detected in a specific cluster >25%.

### 4.4. Single-Cell Trajectory Analysis

Single-cell trajectories were analyzed using a matrix/matrices of cells and gene expression by Monocle (v2.20.0). Monocle reduced the space down to one by two dimensions and ordered the cells. Once the cells were ordered, we could visualize the trajectory in the reduced dimensional space. The trajectory has a tree-like structure, including tips and branches.

### 4.5. Pathway Enrichment Analysis

Gene ontology (GO) and Kyoto Encyclopedia of Genes and Genomes (KEGG) enrichment analysis were used to analyze the function of the differential genes. GO/KEGG bioinformatic analysis was performed using the OmicStudio tools https://www.omicshare.com/ (accessed on 13 August 2023). The calculated *p*-values were false discovery rate (FDR)–corrected, taking FDR ≤0.05 as a threshold.

### 4.6. Cell Culture, Interference, and Induction of Differentiation

The *longissimus dorsi* muscle of the three-day-old piglets was taken, blood vessels and connective tissue were removed, and the muscle tissue was sheared in small pieces and finally cut to mincemeat. The muscle tissue was digested with 2 times the volume of 0.28% collagenase for about 1 h 30 min. Then the digestion was terminated with DMEM/F12 complete medium, filtered and centrifuged to remove erythrocytes, the supernatant was discarded, and the precipitate was washed twice before resuspending the cells in culture medium. The cells were cultured in a diet containing 10% Fetal Bovine Serum (gibco), DMEM/F-12(1:1) basic(1×) (gibco). *THRSP*-siRNA (siRNA was synthesized by Suzhou GenePharma Company (Suzhou, China), the control group was matched with NC, and the concentration was 20 Μm. The *THRSP*-siRNA sequences is: sense (5′-3′): CUGGCCGUGGCUCGCAACATT, antisense (5′-3′): UGUUGCGAGCCACGGCCAGTT). Lipofectamine™ RNAiMAX reagent (purchased from Thermo Fisher Scientific, Waltham, MA, USA) was used when the cell density reached 60–80%. When porcine preadipocytes confluent reached 100% 48 h, lipid formation was induced by classical MDI plus rosiglitazone (DMEM/F12 complete medium: 0.1 mmoL/L 3-isobutyl 1-methylxanthine, 10 mg/L insulin, 1 μmoL/L rosiglitazone, 1 μmoL/L dexamethasone). After 4 d, the maintenance induction medium (50 mL DMEM/F12 complete medium and 142.85 μL insulin) was replaced.

### 4.7. RT-qPCR and BODIPY, Oil-Red-O Staining

After 4 d of inducing preadipocytes adipogenesis according to the previous method, cellular RNA was extracted from preadipocytes using the E.Z.N.A.^®^HP Total RNA Kit (Omega Bio-tek, Shanghai, China) kit. The RNA was reverse transcribed to cDNA using ABScript III RT Master Mix for qPCR with gDNA Remover (ABclonal, RK20429). The cDNA was diluted 1:5 to detect cellular RNA expression using 2× Universal SYBR Green Fast qPCR Mix (ABclonal, RK21203) (the sequence of internal reference and target gene primers is shown in Appendix A). In order to observe the formation of the lipid droplets, the lipid droplets were stained with BODIPY (Thermo Fisher) and Oil-RED-O (Improved Oil-red-O staining kit, Biyun Tian) eight days after induction, according to the instructions in the kit.

## 5. Conclusions

In summary, according to our knowledge, this is the first description of cellular heterogeneity of porcine skeletal muscle at two developmental stages in Xidu black pigs. In addition, we have identified the *THRSP* gene as a biomarker for porcine IMF cells. Our results could provide a theoretical basis for the study of pig breeding and human muscle diseases.

## Figures and Tables

**Figure 1 ijms-25-12935-f001:**
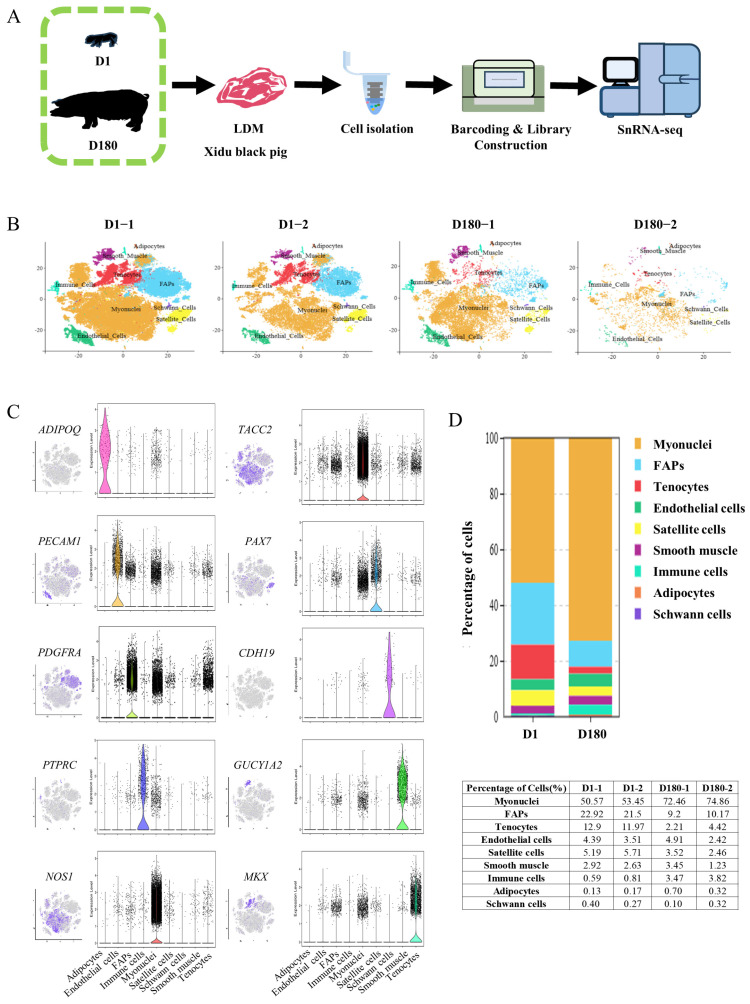
SnRNA-seq identifies distinct cell populations in the LDM of the Xidu black pig D1 and D180 groups. (**A**) Scheme of the experimental design for snRNA-seq on LDM nuclei. (**B**) tSNE visualization of all of the isolated single nuclei from D1 and D180 muscle coloured by cluster identity. Each dot represents a cell, and the color of the dot is related to the amount of gene expression selected. (**C**) t-SNE and violin plot displaying the expression of selected marker genes for each cluster of nuclei. (**D**) Nuclear proportion in each cluster in D1 and D180 muscles. Each cluster is colour-coded.

**Figure 2 ijms-25-12935-f002:**
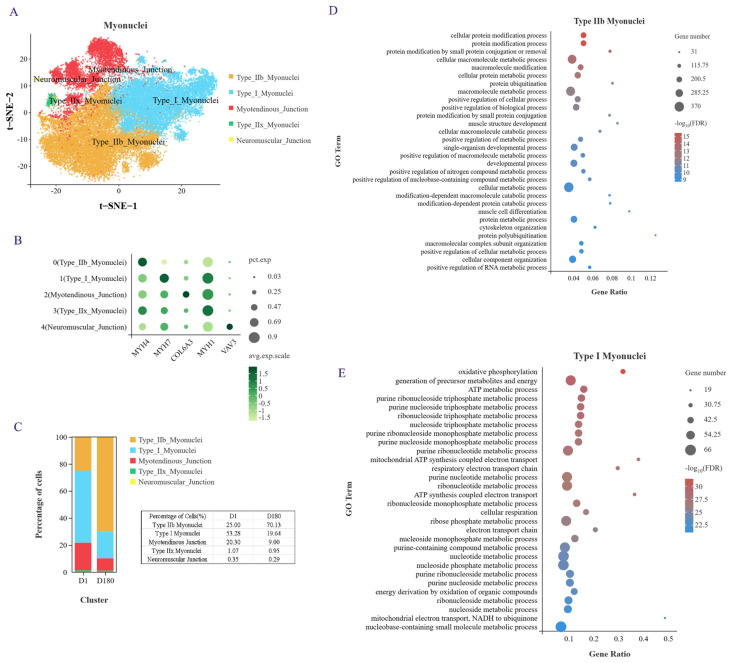
Heterogeneity analysis of myonuclei subpopulation. (**A**) myonuclei were divided into 5 subgroups and t-SNE visualized. (**B**) Specific expression genes of 5 subgroups of myonuclei. (**C**) The proportion of each cell subpopulation. (**D**) GOBP enrichment analysis of genes in type IIb myonuclei. (**E**) GOBP enrichment analysis of genes in type I myonuclei.

**Figure 3 ijms-25-12935-f003:**
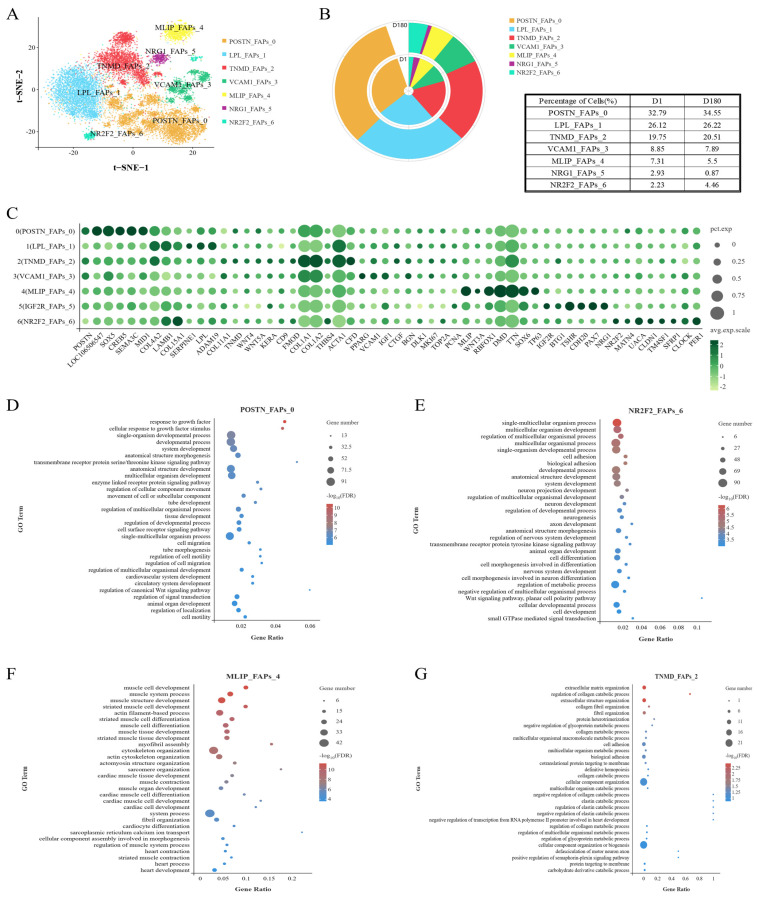
Swine FAPs are a heterogeneous population. (**A**) FAPs were divided into seven subgroups and t-SNE visualized. (**B**) Ring charts and tables for each cell subpopulation on D1 and D180. (**C**) Specifically expressed genes in seven subgroups of FAP. (**D**) GOBP enrichment analysis of genes in FAP 0. (**E**) GOBP enrichment analysis of genes in FAP 6. (**F**) GOBP enrichment analysis of genes in FAP 4. (**G**) GOBP enrichment analysis of genes in FAP 2.

**Figure 4 ijms-25-12935-f004:**
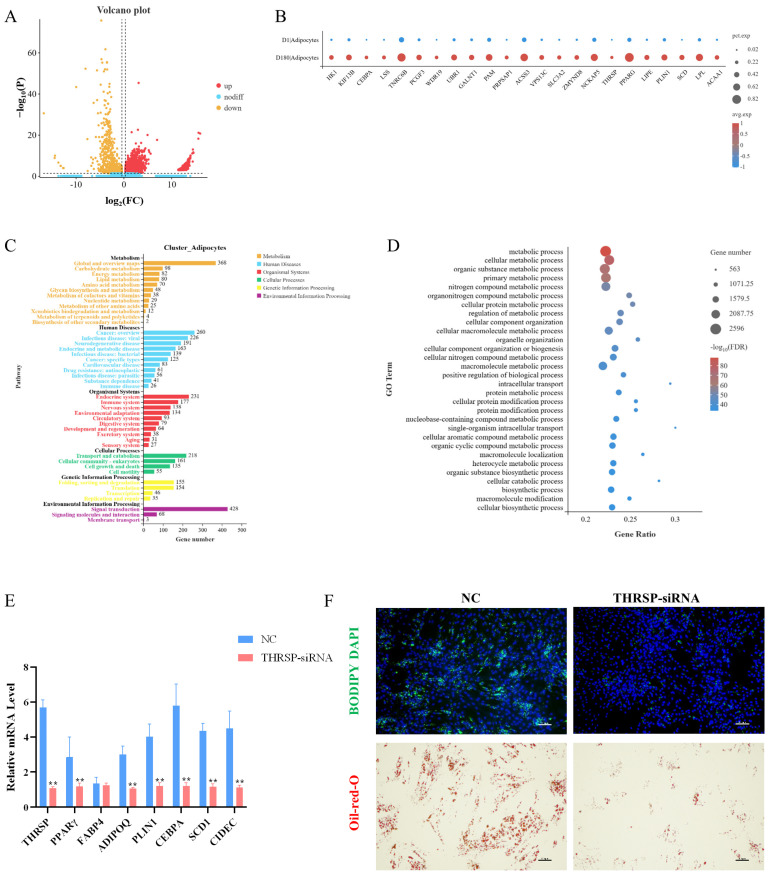
(**A**) Volcano map of differential genes between groups D1 and D180 in adipocyte populations. (**B**) Specificity of differential gene expression in D1 and D180 groups. The size of the bubble represents the proportion of genes expressed in the subpopulation, and the larger the bubble, the higher the proportion. The bubble color represents the normalized value of the average gene expression in the subpopulation. (**C**) Enrichment of KEGG pathway in adipocyte subsets. (**D**) Enrichment of GOBP in adipocyte subsets. (**E**) Preadipocytes were transfected with *THRSP*-siRNA and NC, and the expression levels of *THRSP*, *PPARγ*, *FABP4*, *ADIPOQ*, *PLIN1*, *CEBPA*, *SCD1,* and *CIDEC* were detected by real-time fluorescence quantification PCR(RT-qPCR) 4 days after transduction. ** means *p* < 0.01. (**F**) BODIPY staining and oil red O staining showed the effect of *THRSP*-siRNA on the fat forming capacity of preadipocytes, respectively. Green and red indicate lipid droplets, and blue indicates the nucleus. Each group is n = 3.

**Figure 5 ijms-25-12935-f005:**
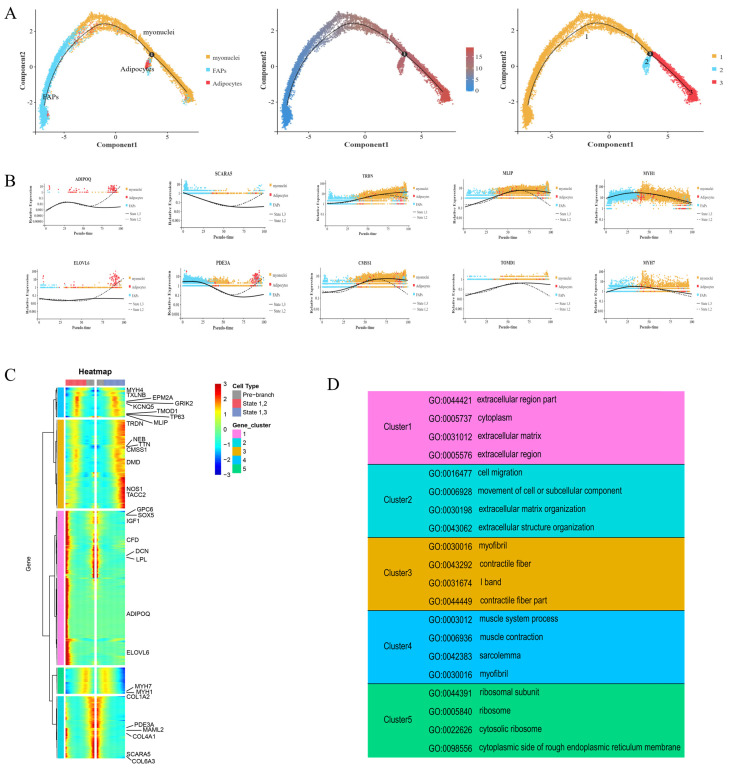
Differential gene expression profiles during FAP cell differentiation. (**A**) The pseudotime analysis result of FAPs, adipocytes, and myocytes. Each dot represents one nucleus, and each branch represents one cell state. (**B**) Scatter plot of the expression levels of representative characteristic genes across pseudotime. (**C**) Five clusters of pseudotime gene expressions are clustered hierarchically during FAP cell differentiation. (**D**) The representative gene functions and pathways of each cluster.

**Table 1 ijms-25-12935-t001:** The results obtained from Cell Ranger analyses.

Sample	Estimated Number of Nuclei	Mean Reads Per Nuclei	Median Genes Per Nuclei	Total Genes Detected	Median UMI Counts Per Nuclei
D1_1	27,560	14,121	1150	22,556	2059
D1_2	27,677	13,028	1033	22,446	1748
D180_1	19,842	22,662	1194	22,547	2399
D180_2	3223	128,122	1391	21,808	2910

**Table 2 ijms-25-12935-t002:** Nuclei number of snRNA-seq datasets before or after filter from each group.

Groups	Nuclei Number after Filter	Median UMI Counts Per Nuclei After Filter	Median Genes Per Nuclei After Filter
D1	43,586	1716.5	994
D180	19,685	2163	1115

## Data Availability

Raw data can be obtained by contacting the corresponding author.

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
