# Peer review of "A Single-Cell Atlas of Porcine Skeletal Muscle Reveals Mechanisms That Regulate Intramuscular Adipogenesis"

_ijms, 2024, doi:10.3390/ijms252312935_

Round 1
Reviewer 1 Report
Comments and Suggestions for Authors
The authors present an interesting basic research study. It is mainly well described, but it should be clarified that nuclei were analyzed and not cells and the description should be more precise. Some more details of used methods should be given. Thus, the manuscript should be revised before it can be published.
What are “myonuclei cells”? Where appropriate it could be described as nuclei of muscle fibers, adipocytes, FAP, etc.
Specific comments
L14 animal species and breed should be mentioned
L20 “cells” should be “nuclei”, because several myonuclei belong to one muscle fiber
L26 “biomarkers” should be “biomarker”
L27 “significantly” should be “significant”
L30 “quality” of what?
L69 is it correct to say “via high-throughput single-cell transcriptome sequencing”?
L81, 82, 94ff and elsewhere be more precise in the description, when the term “cells” or “per cell” is used, but nuclei were analyzed
L90 isn’t it better to use “muscle fibers” instead of “myonuclei-containing cells”?
Table 1 explain the discrepancy in the estimated number of cells and reads between the two 180d pigs. Is it the number of genes or rather transcripts that is reported?
L108 add the animal species and breed
L138 did you check the muscle fiber types in histological muscle sections for confirmation?
L141-148 needs clarification
L162 but 7 clusters are described
L175 and others: not the cells but the expressed genes are “enriched”
L189 please explain: “involved in the negative 189 regulation of IMF”
L244-250 since the C3H10 is a mouse cell line, the gene names should be written with capital and lowercase letters as in the figure.
L267 add the version and company of the software
L335 suggest to add “transcriptome of” after “detect”
L380-387 it should be discussed that THRSP was shown a marker of adipocyte number in muscle of cattle and not of higher fat deposition activity (Schering et al. IJBS 2017). It may be comparable in porcine muscle.
A more detailed description of nuclei preparation must be added.
L443-449 rewrite to clarify the description. When were the cells harvested for analysis?
L450-454 more details should be added. The primer list in Table S1 does not contain a reference gene primer.
L460 suggest to add “cellular” before “heterogeneity”
L465 information on supplementary material must be added
L478-484 Where are the data sets deposited?
Reviewer 2 Report
Comments and Suggestions for Authors
Xu and colleague compared the comprehensive gene expression profiles of porcine skeletal muscle between Day 1 and 180. Consistent with the accumulation of intramuscular fat, adipocytes were increased in Day 180 group. They showed that FAPs are the origin of adipocytes and classified into 6 population based on the gene expression and their deduced functions. It was also suggeste that the THRSP gene is a biomarker for intramuscular fat cells involved in lipid droplet formation. This study contains many interesting insights in the living stock industry. However, some revision is necessary.
1. Fig.3: SCARA5 is not a proper marker for FAPs, because its expression is committed the cells into adipocytes. It is probable that these populations are adipocyte progenitors, not FAPs. Why the authors did not use PDGRFalpha as a maker?
2. The authors must discuss about the relationship of original MSCs and identified 6 FAP populations.
3. Fig. 4 E and F: Rodent THRSP gene is already known to be involved in lipogenesis. The authors should use porcine MSCs in this experiment.
4. 8 days were necessary to induce differentiation of MSCs into adipocytes. THRSP expression was inhibited by siRNA for such long time. Fig 4E is the results of Day2.
5. Discussion section is too short. In addition to the comment 2, the authors must add the discussion about other results.
Round 2
Reviewer 2 Report
Comments and Suggestions for Authors
The authors tackled to all my points.